# Possible Applications for a Biodegradable Magnesium Membrane in Alveolar Ridge Augmentation–Retrospective Case Report with Two Years of Follow-Up

**DOI:** 10.3390/medicina59101698

**Published:** 2023-09-22

**Authors:** Daniel Palkovics, Patrick Rider, Svenja Rogge, Željka Perić Kačarević, Peter Windisch

**Affiliations:** 1Department of Periodontology, Semmelweis University, Szentkirályi Utca 47, 1088 Budapest, Hungary; peter.windisch@gmail.com; 2Botiss Medical AG, Ullsteinstraße 108, 12109 Berlin, Germany; patrick.rider@botiss.com (P.R.); svenja.rogge@botiss.com (S.R.); zeljka.peric.kacarevic@botiss.com (Ž.P.K.); 3Department of Anatomy Histology and Embryology, Faculty of Dental Medicine and Health, Josip Juraj Strossmayer University, Ul. Cara Hadrijana 8/A, 31000 Osijek, Croatia

**Keywords:** magnesium membrane, resorbable barrier membrane, staged GBR, 3D evaluation, CBCT subtraction

## Abstract

*Background and Objectives*: A rigid, resorbable magnesium membrane was recently developed, combining the advantages of resorbable and non-resorbable membranes. Our aim was to describe the application of this membrane for guided bone regeneration (GBR). *Materials and Methods*: This case report described the treatment and 3D evaluation of two cases utilizing a resorbable magnesium barrier membrane. In Case #1, GBR was performed with a bilayer tunnel flap. The magnesium barrier was placed fixed subperiosteally through remote vertical incisions. In Case #2, GBR was performed using a split-thickness flap design. Volumetric and linear hard tissue alterations were assessed by 3D cone-beam computed tomography subtraction analysis, as well as with conventional intraoral radiography. *Results*: Case #1 showed a volumetric hard tissue gain of 0.12 cm^3^, whereas Case #2 presented a 0.36 cm^3^ hard tissue gain. No marginal peri-implant hard tissue loss could be detected at the two-year follow-up. *Conclusions*: The application of conventional resorbable collagen membranes would be difficult in either of the cases presented. However, the rigid structure of the magnesium membrane allowed for the limitations of conventional resorbable membranes to be overcome.

## 1. Introduction

Modern surgical approaches for alveolar ridge augmentation have allowed for dental implant placements in edentulous areas with advanced alveolar bone deficiencies [1]. Of the various alveolar ridge augmentation procedures, guided bone regeneration (GBR) and onlay grafting are among the most effective for rehabilitating alveolar ridge defects [2]. GBR is often prioritized by patients and clinicians, as it is a less invasive procedure that does not require the preparation of a second intra- or extraoral donor site.

Resorbable or non-resorbable barrier membranes can be used in GBR procedures, with each membrane type having specific advantages and shortcomings. Resorbable membranes do not require a second surgical intervention for their removal; however, they lack structural rigidity and are less suitable for the reconstruction of large alveolar ridge defects [3]. In contrast, non-resorbable membranes (e.g., titanium-reinforced polytetrafluoroethylene (PTFE) membranes, and titanium meshes) have excellent shape-maintaining capabilities. However, their greatest shortcoming involves the necessity for a re-entry surgery or the preparation of a larger flap during the implant placement. In terms of membrane exposure, complications are generally easier to control in conjunction with resorbable membranes [4,5,6], although a recent publication demonstrated that the event of a membrane exposure, utilizing a non-reinforced PTFE membrane did not compromise the clinical outcome [7].

A resorbable membrane made of a magnesium alloy (NovaMag^®^, botiss GmbH, Zossen, Germany) has recently been introduced that combines the advantageous properties of both resorbable and non-resorbable membranes [8]. This new membrane has a similar structural rigidity to a titanium mesh; however, it has been shown to completely resorb until the 16th postoperative week [9]. During the degradation process, the membrane is gradually converted into magnesium salts while hydrogen gas is released [10,11]. It was demonstrated that the salts are completely resorbed by the body, while the cavities formed by the gas release spontaneously resolved and had no negative effect on bone regeneration [8,9,10,12]. A previous animal study in beagle dogs compared the performance of this new magnesium membrane to the gold standard collagen membrane (Bio-Guide^®^, Geistlich Pharma, Wolhusen, Switzerland). The authors observed similar performances of the two membranes in terms of hard tissue gain and membrane degradation [9]. As a result, the study concluded that the magnesium membrane was suitable for GBR treatments. More recently, Elad et al. utilized the resorbable magnesium barrier to establish a lost buccal bone wall in conjunction with an immediate implant placement protocol. In this case report none of the patients presented severe postoperative complications [13].

The aim of this case report was to present two possible surgical approaches for GBR using the magnesium barrier membrane and to evaluate the efficacy of the membrane by performing volumetric and linear radiographic measurements. The additional aim was to investigate long-term clinical outcomes.

## 2. Materials and Methods

### 2.1. Patient Selection

The current case report presented the retrospective evaluation of two cases treated with resorbable magnesium barrier membranes for GBR. The CARE guidelines were followed [14], and the corresponding CARE checklist is attached as a supplementary file (Appendix A). The investigation was conducted in full accordance with the 2013 revision of the Declaration of Helsinki [15]. Both patients confirmed their understanding of the surgical procedure and provided written informed consents for both the procedure and subsequent publication. Neither participant had relevant medical conditions that contraindicated the augmentation procedure. Both participants had good oral hygiene (full mouth plaque score <20%) and compliance.

### 2.2. Radiographic Image Acquisition

Cone-beam computed tomography (CBCT) scans were taken before and six months after the augmentation procedures using an I-CAT FLX^®^ (KaVo Dental GmbH, Bieberach an de Riß, Germany) CBCT machine with the following parameters: 300 μm voxel size, 120 kV anode voltage, and 36 mA X-ray tube current [16].

### 2.3. Surgical Treatment

#### 2.3.1. Case #1

The patient was 38 years old and had lost her lower right central incisor due to advanced periodontal bone loss. The coronal portion of Tooth 41 was splinted to the neighboring teeth with a temporary fiber-reinforced composite splint. Tooth 31 presented a Miller Class III gingival recession. The defect was classified as a medium horizontal alveolar ridge defect according to the horizontal-vertical-combination (HVC) classification system [17]; therefore, a staged horizontal alveolar ridge augmentation and implant placement were scheduled.

The alveolar ridge augmentation was performed with a bilayer tunnel flap, similar to the extraction site development technique [18,19]. After local anesthesia, mesial and distal remote vertical incisions were prepared one interdental space away from the surgical area at the buccal aspect. The superperiosteal tunnel was prepared by sharp dissection with tunneling knives (Tunneling Knives^®^, Deppler, Rolle, Switzerland). The periosteum was then bluntly dissected from the bone to create the subperiosteal tunnel. Prior to insertion, the magnesium barrier was shaped with special magnesium scissors (NovaMag^®^ scissors). The sharp edges of the membrane were flattened and bent into shape using a NovaMag^®^ sculptor instrument (Figure 1A).

The magnesium membrane was placed under the periosteum and fixed with anodized titanium pins to maintain a safe secluded space between the membrane and the bone surface. This space was filled with a 1:1 mixture of locally harvested autogenous bone chips and a bovine-derived xenograft (BDX) (cerabone^®^, botiss GmbH, Zossen, Germany) (Figure 1B). A xenogenic collagen matrix (Mucoderm^®^, botiss, Zossen, Germany) was fixed submucosally to achieve an ideal soft tissue contour. Remote vertical incisions were sutured with horizontal mattress sutures using 5–0 monofilament suturing material (Dafilon, B Braun^®^, Melsungen, Germany). Sutures were removed 14 days postoperatively.

Following a six-month healing period, a 2.9 mm diameter sandblasted, large-grit, acid-etched implant was placed without any further augmentation [20,21,22] (Appendix A).

#### 2.3.2. Case #2

A 56-year-old patient was referred for alveolar ridge augmentation in the lower lateral region. The patient had an edentulous area in the position of Teeth 45 and 46. The alveolar ridge defect was classified as a medium combination defect according to the HVC classification, and the patient was scheduled for staged alveolar ridge augmentation and implant placement. In contrast to the previous case, a tunnel-like flap was not feasible due to the advanced morphology of the defect. Therefore, a more conventional split-thickness flap design was used to access the surgical area [23]. Under local anesthesia, a midcrestal incision was me that extended to the gingival sulcus of the neighboring teeth on the buccal aspect. Subsequently, a full-thickness preparation was performed until the mucogingival junction. Thereafter, partial thickness flap elevation was conducted with the use of surgical blades and tunneling instruments (Tunneling Knives^®^, Deppler, Rolle, Switzerland) (Figure 2A). After the separation of the mucosal layer, the periosteum was elevated from the bone surface by blunt dissection (Figure 2B). Autogenous bone chips were harvested locally and mixed with BDX in a sterile Petri dish in a 1:1 ratio (Figure 2C). Prior to insertion, the magnesium barrier was shaped with special magnesium scissors (NovaMag^®^ scissors) (Figure 2D), and the sharp edges of the membrane were flattened and bent into shape using a NovaMag^®^ sculptor instrument. The membrane was fixed with anodized titanium pins on the lingual aspect (Figure 2E), after which the graft was compacted tightly, and the membrane was folded over the grafted area and secured on the buccal aspect (Figure 2F). The periosteal layer was sutured with horizontal mattress sutures to the lingual full-thickness flap using 4–0 non-resorbable monofilament sutures (Figure 2G). The buccal mucosal layer was then sutured to the lingual flap with horizontal mattress sutures using 5–0 non-resorbable monofilament suturing material (Figure 2H). The sutures were removed after 14 days.

Following a six-month healing period, the patient was scheduled for implant placement (Appendix A). Subsequently two implants were placed without any further augmentation.

### 2.4. 3D Radiographic Evaluation

Baseline and six-month follow-up CBCT scans were imported into an open-source Digital Imaging and Communications in Medicine (DICOM) imaging software platform (3D Slicer 5.3.0., www.slicer.org, 31 January 2023) [24,25]. Similar to procedures used in other medical fields [26,27] and those of previous studies [28], a semi-automatic image segmentation method was used to acquire 3D models of the pre- and post-operative CBCT scans. Following segmentation, the pre- and post-operative CBCT scans were spatially registered using an automatic voxel intensity-based registration method (3D Slicer, elastix extension) [29]. To visualize and calculate the hard tissue gain, the preoperative 3D models were subtracted from the postoperative 3D models [7,30], and the volumetric differences were calculated and expressed in cubic centimeters (cm^3^). In addition, horizontal and vertical linear hard tissue changes were recorded at the deepest point of the defect. Vertical new hard tissue formations were measured midcrestally, parallel to the long axis of the alveolar ridge, whereas horizontal hard tissue changes were measured perpendicularly to the long axis of the alveolar ridge, at 1, 2, and 3 mm apical to the top of the alveolar ridge.

### 2.5. Long-Term Radiographic Evaluation

Conventional intraoral radiographs (IRs) were obtained before and directly after implant placement using a parallel long cone technique. After implant placement and the delivery of the final prosthetic restoration, patients were regularly recalled and control Irs were acquired, with the final IR taken two years after implant placement in both cases.

### 2.6. Outcome Measures

The primary outcome of the investigation was to evaluate the radiographic volumetric hard tissue gain following alveolar ridge augmentation with the resorbable magnesium membrane. Secondary outcomes were: (i) evaluations of linear hard tissue changes, (ii) assessments of 3D morphological alterations, and (iii) evaluations of long-term radiographic results.

## 3. Results

### 3.1. Case #1

#### 3.1.1. Baseline Clinical Situation

The edentulous alveolar ridge at baseline was 2.05 mm, 3.24 mm, and 3.67 mm wide, respectively, at 1, 2, and 3 mm-s apical from the top of the edentulous crest. The defect was classified as a medium horizontal alveolar ridge defect. Tooth 31 presented a 4.51 mm deep buccal bone dehiscence.

#### 3.1.2. Short-Term Hard Tissue Gain and Two-Year Follow-Up

Following spatial alignment and 3D subtraction, a volumetric hard tissue gain of 0.12 cm^3^ was detected (Figure 3A,B). At the deepest point of the defect, a 0.79 mm vertical hard tissue gain was measured, while horizontal hard tissue gains of 0.93 mm, 1.23 mm, and 1.38 mm were observed at 1, 2, and 3 mm-s apical to the top of the alveolar crest, respectively (Figure 3C,D). The initially measured dehiscence at Tooth 31 was reduced to 2.68 mm (Figure 4).

No marginal bone loss was detected in the long term when comparing the IR after the delivery of the final restoration to the IR of the two-year control, although increased corticalization and slight coronal hard tissue creeping were detected (Figure 5). The data are summarized in Appendix A.

### 3.2. Case #2

#### 3.2.1. Baseline Clinical Situation

The edentulous alveolar ridge at baseline was 1.74 mm, 3.34 mm, and 5.28 mm wide at 1, 2, and 3 mm-s apical from the top of the edentulous crest, respectively. The defect was classified as a medium combination alveolar ridge defect according to the HVC classification system.

#### 3.2.2. Short-Term Volumetric Hard Tissue Gain and Long-Term Follow-Up

Compared to the baseline CBCT scan, a 0.36 cm^3^ hard tissue gain could be detected after the six-month healing period (Figure 6A,B). A 2.77 mm vertical hard tissue gain was detected at the deepest point of the defect, while horizontal hard tissue gains were 4.37 mm, 4.33 mm, and 3.37 mm at 1, 2, and 3 mm apical to the top of the alveolar crest, respectively (Figure 6C,D).

At the two-year follow-up, the marginal bone was stable, and no resorption was visible (Figure 7). The data are summarized in Appendix A.

## 4. Discussion

Various types of membranes can be used for alveolar hard tissue reconstruction, although large defects require barriers that can maintain their shape and provide a safe secluded space. Titanium-reinforced PTFE membranes and titanium meshes are considered the gold standards for the reconstruction of large alveolar ridge deficiencies [31]. However, these types of barrier membranes must be removed during separate surgical interventions, and they are more likely to become exposed during the healing period due to their non-resorbable properties [5,31]. Information on the exposure rate of the magnesium barrier membrane in humans is currently lacking; however, a previous animal study demonstrated that if an exposure occurred the dehiscence resolved spontaneously 10 days after exposure [8,9]. The release of hydrogen gas was initially suggested to interfere with the wound healing processes, although neither the animal studies [8,9,13] nor the current clinical case report found any negative effects from the gas release. One of the patients reported a metallic taste after the surgery, although it spontaneously resided, and the healing was uneventful.

The two cases presented in the current paper did not allow conventional resorbable collagen membranes to be used; however, the limitations of these membranes could be overcome with the NovaMag^®^ membrane because of its rigid structure.

Similar to the study by Elad et al. [13], the first case in the current paper used the magnesium membrane as a wall on the buccal aspect in conjunction with a bilayer tunnel approach. By pre-bending the membrane, the ideal curvature of the alveolar ridge in the lower anterior region could be re-established. While collagen barrier membranes are generally difficult to manipulate when combined with a tunnel approach, the rigidity of the NovaMag^®^ promotes this material as an ideal alternative. The hard tissue gain was observed at the edentulous crest and a partial resolution of the buccal bone dehiscence was evident at the adjacent tooth. This resulted in soft tissue coverage of the Miller Type III gingival recession.

The second presented case used the magnesium membrane with a more conventional approach. By elevating a split-thickness flap on the buccal aspect, horizonto-vertical hard tissue augmentation occurred. In addition, due to the rigidness of the membrane, no tenting screws were necessary to support the barrier from below [7]. Volumetric hard tissue gain was found to be similar to results acquired with PTFE membranes [7,23] or titanium meshes [6].

The current study has reported two-year follow-ups using the resorbable magnesium barrier membrane. The IRs acquired after the delivery of the final prosthetic restoration were compared to those at the two-year follow-up and stable marginal bone contours were observed. In both cases, a stable peri-implant hard and soft tissue environment was established, and neither of the inserted implants showed marginal bone resorption or peri-implant inflammatory processes.

The aim of this pilot case report was to introduce concepts for the application of the resorbable rigid magnesium barrier membrane.

## 5. Conclusions

Two clinical applications of the NovaMag^®^ membrane were presented in this case report that verified the diversity of this material. Although definitive conclusions cannot be derived from the results of two cases, the clinical and radiographic results are promising, since successful results could be obtained both in the short and long term, as was demonstrated in this study. Although the two cases were treated differently, both clarified the advantages of the rigid membrane structure. Large sample prospective controlled clinical trials are necessary to validate the efficacy of the membrane and understand the potential disadvantages of this material.

## Figures and Tables

**Figure 1 medicina-59-01698-f001:**
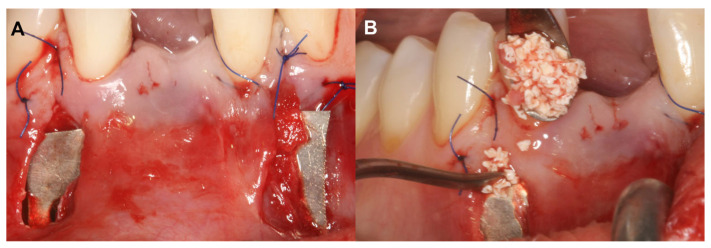
Bilayer tunnel approach for guided bone regeneration (Case #1). (**A**): Resorbable magnesium (NovaMag^®^) membrane positioned under the periosteum, Xenogeneic collagen matrix (mucoderm^®^) fixed between the mucosa and the periosteum. Application of xenogeneic collagen matrix was necessary to establish a proper soft tissue contour at implantation site and to reduce the gingival recession at tooth 31; (**B**): 1:1 ratio composite graft placed between the magnesium barrier and the buccal cortical plate.

**Figure 2 medicina-59-01698-f002:**
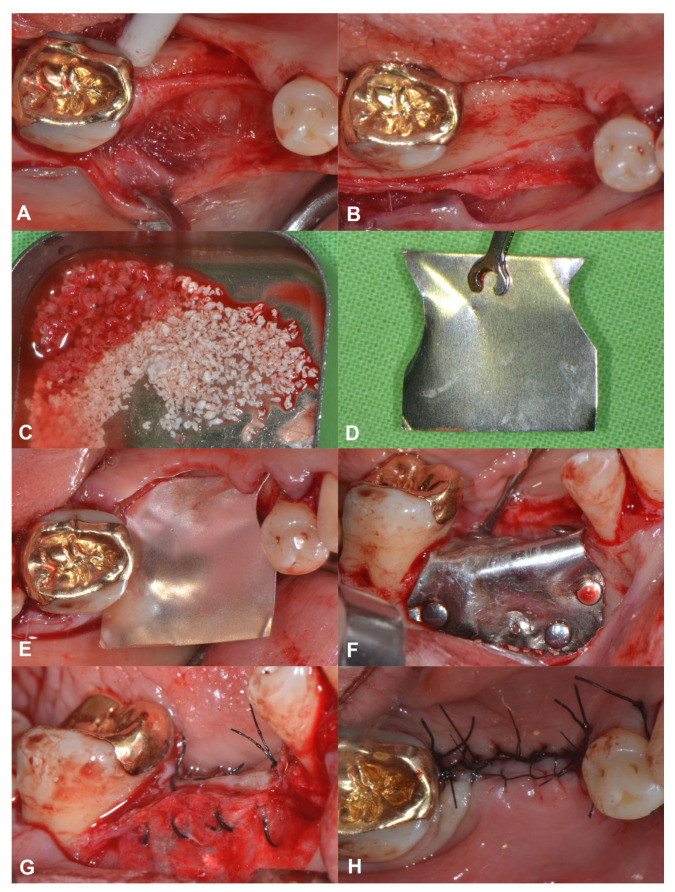
Horizonto-vertical guided bone regeneration with a split-thickness flap design (Case #2). (**A**): Preparation of the mucosal layer; (**B**): Preparation of the periosteal layer; (**C**): Composite graft: 50% autogenous bone chips–50% bovine derived xenograft; (**D**): Shaping the magnesium barrier membrane; (**E**): Positioning of the magnesium barrier membrane; (**F**): Fixation of the magnesium barrier membrane; (**G**): Suturing of the periosteal layer with 4–0 monofilament sutures; (**H**): Suturing of the mucosal layer with 5–0 monofilament sutures.

**Figure 3 medicina-59-01698-f003:**
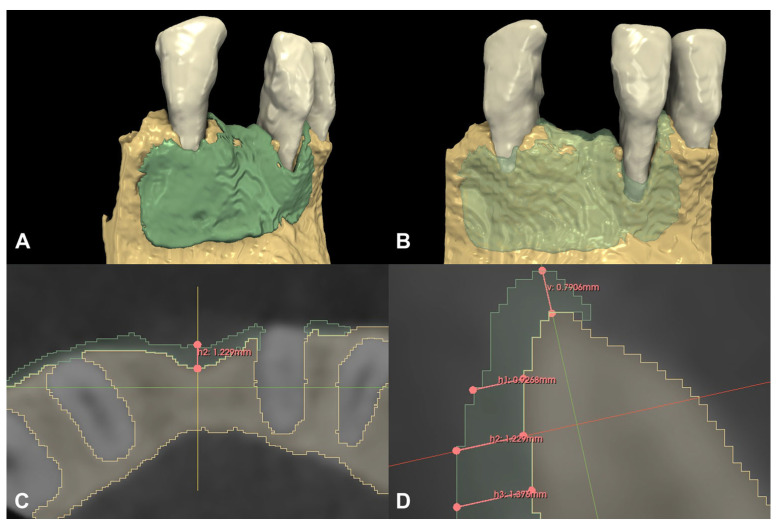
Volumetric and linear radiographic evaluation of Case #1. (**A,B**): Volumetric- and 3D morphological hard tissue alterations; (**C**): Linear hard tissue gain at 2 mm apical to the edentulous crest (H2) (axial view); (**D**): Linear hard tissue changes (sagittal view) (V: vertical change, measured midcrestally parallel to the long axis of the edentulous ridge, H1: horizontal change measured perpendicularly to the long axis of the edentulous ridge 1 mm apical to the alveolar crest (H1 = 0.9268 mm), H2: horizontal change measured perpendicularly to the long axis of the edentulous ridge 2 mm apical to the alveolar crest (H2 = 1.229 mm), H3: horizontal change measured perpendicularly to the long axis of the edentulous ridge 3 mm apical to the alveolar crest (H3 = 1.376 mm).

**Figure 4 medicina-59-01698-f004:**
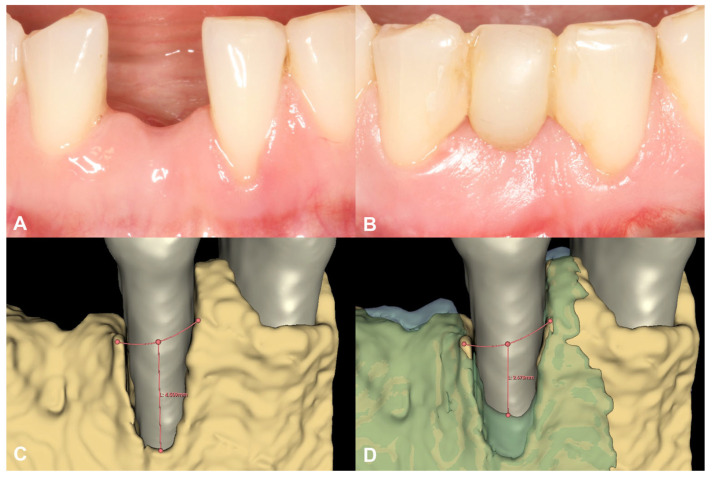
Soft tissue conditions and radiographic hard tissue changes around Tooth 31 (Case #1). (**A**): Soft tissue conditions at baseline, Miller Class III recession at Tooth 31; (**B**): Soft tissue conditions at six-month follow-up, resolution of the recession at Tooth 31; (**C**): Baseline buccal alveolar bone dehiscence measured at 4.51 mm at Tooth 31; (**D**): Partial resolution of the buccal alveolar bone dehiscence at Tooth 31, residual depth: 2.68 mm.

**Figure 5 medicina-59-01698-f005:**
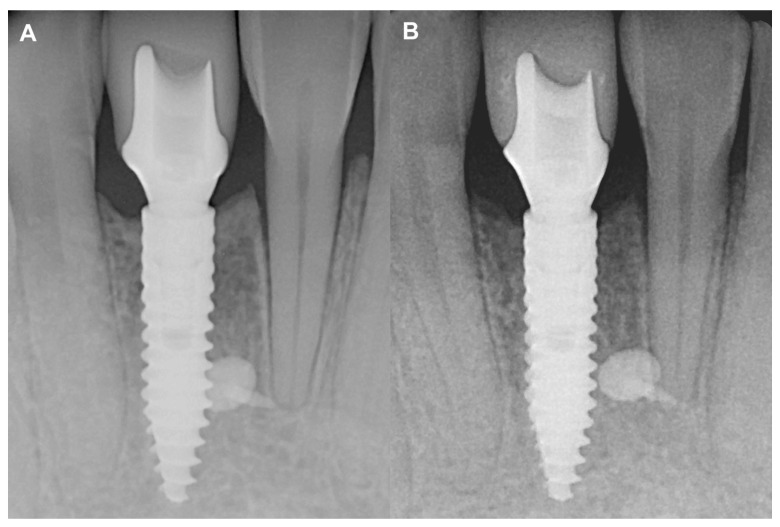
Two-year follow-up of Case #1. (**A**): Intraoral radiograph at the delivery of the final restoration; (**B**): Intraoral radiograph at two-year follow-up.

**Figure 6 medicina-59-01698-f006:**
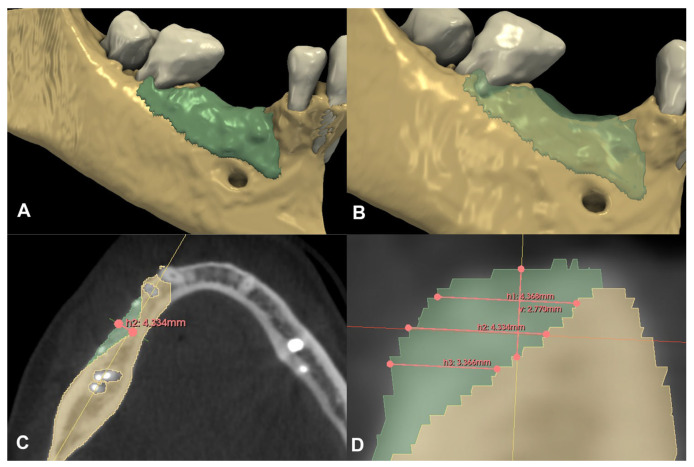
Volumetric and linear radiographic evaluation of Case #2. (**A**,**B**): Volumetric- and 3D morphological hard tissue alterations; (**C**): Linear hard tissue gain at 2 mm apical to the edentulous crest (H2) (axial view); (**D**): Linear hard tissue changes (sagittal view) (V: vertical change, measured midcrestally parallel to the long axis of the edentulous ridge, H1: horizontal change measured perpendicularly to the long axis of the edentulous ridge 1 mm apical to the alveolar crest, H2: horizontal change measured perpendicularly to the long axis of the edentulous ridge 2 mm apical to the alveolar crest, H3: horizontal change measured perpendicularly to the long axis of the edentulous ridge 3 mm apical to the alveolar crest).

**Figure 7 medicina-59-01698-f007:**
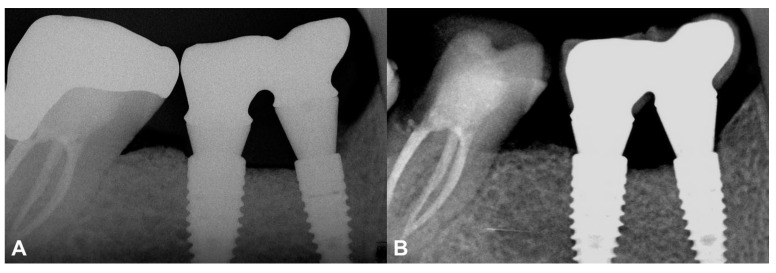
Two-year follow-up of Case #2. (**A**): Intraoral radiograph at the delivery of the final restoration; (**B**): Intraoral radiograph at two-year follow-up.

## Data Availability

All data that were used for this study are included in the paper or the Appendix A.

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
