# Peer review of "Possible Applications for a Biodegradable Magnesium Membrane in Alveolar Ridge Augmentation–Retrospective Case Report with Two Years of Follow-Up"

_medicina, 2023, doi:10.3390/medicina59101698_

Round 1
Reviewer 1 Report
Thanks to the authors for the study. In the study, two surgical possibilities using a magnesium barrier membrane for guided bone regeneration are presented. The study is unique in that it addresses a different possibility for directed bone regeneration. It is important in terms of presenting the results of using magnesium membrane for guided bone regeneration according to previous studies. The methodology for cases and surgical procedure has been presented in an appropriate manner. The results are favorable and indeed a larger sample size and follow-up would be required for clinical generalizability. But for a case report, the results are appropriate. The discussion section needs to be expanded a little more by associating/comparing with current studies. References are appropriate but should be arranged according to the journal template. Figures are appropriate.Minor editing of English language required.
Author Response
Dear Reviewer,
Thank you for your positive response and the constructive suggestions to our article. The discussion section was change slightly, however not to many studies with magnesium reported similar outcomes to ours. The reference output style was changed to the style of the journal.
Reviewer 2 Report
The application of conventional resorbable collagen membranes would be difficult in either of the cases presented. Some comments given as follows:
1. Line 37, It is unclear whether the author's something new in this work. According to the evaluation, several published literature by other researchers in the past adequately explain the issues you made in the present paper. Please be careful to highlight in the introduction section anything really innovative in this work.
2. Line 74-75, please develop the objective of present study since it is too simple.
3. Line 77, Given that the current form is inappropriate, the authors must address the basis for patient selection. Is any standard, procedure, or protocol used? The included patients are extremely small and heterogeneous, and there is no true group control. As it is one of the main problems with the current submissions and the justification for suggesting a rejection, the reviewer kindly ask that you take this matter seriously.
4. Line 122, please make a comprehensive discussion for bilayer tunnel approach in this illustration.
5. Please explain the urgency of biocompability aspect of biomaterials used. It is important to ensure there is no negative response trough the body after implantation. Explain this crucial point along with relevant reference as follows: https://doi.org/10.3390/jcs7080324, https://doi.org/10.3390/ma14247554, https://doi.org/10.3390/ma16124458
-
Author Response
Dear Reviewer,
Thank you for your review of our article, please find our point-by-point response to your concerns.
The application of conventional resorbable collagen membranes would be difficult in either of the cases presented.
Indeed application of resorbable collagen membranes would be very difficult in either case. That is why these two cases were presented to show that the magnesium membrane can overcome the limitations of materials used so far.
Line 37, It is unclear whether the author's something new in this work. According to the evaluation, several published literature by other researchers in the past adequately explain the issues you made in the present paper. Please be careful to highlight in the introduction section anything really innovative in this work.
You are right that there already were case report articles that reported the use of magnesium in a GBR setting. However, none of the article provided sufficient evaluation of the results (linear and volumetric CBCT measurements) nor did any article present 2-year follow-up data. According to your suggestions sections in the text were rephrased.
Line 74-75, please develop the objective of present study since it is too simple.
“Aim of the study” section was expanded according to reviewers suggestion
Line 77, Given that the current form is inappropriate, the authors must address the basis for patient selection. Is any standard, procedure, or protocol used? The included patients are extremely small and heterogeneous, and there is no true group control. As it is one of the main problems with the current submissions and the justification for suggesting a rejection, the reviewer kindly ask that you take this matter seriously.
The current study is a case report of two cases, therefore a control group is not feasible. As it was also described in the manuscript these two entirely different cases were selected to demonstrate the diversity of the material and to suggest possible future applications for it.
Line 122, please make a comprehensive discussion for bilayer tunnel approach in this illustration.
Description of Figure 1 was enlarged
Please explain the urgency of biocompability aspect of biomaterials used. It is important to ensure there is no negative response trough the body after implantation. Explain this crucial point along with relevant reference as follows: https://doi.org/10.3390/jcs7080324, https://doi.org/10.3390/ma14247554, https://doi.org/10.3390/ma16124458
Biocompatibility of this material has been previously examined and reported thoroughly by previous animal studies (cited in the article). Although biocompatibility is a serious issue it is not the focus of the current article.
Additionally, reviewer has raised concerns regarding the use of English language. Prior to submission the manuscript was sent to proof reading (Cambridge Proof reading sevices). However, the text was reviewed one more time after reviewer suggestion.
Round 2
Reviewer 1 Report
The authors addressed adequately to the suggestions.
Minor editing of English language required.